# ViT-EBoT: Vision Transformer for Encrypted Botnet Detection in Resource-Constrained Edge Devices

## Abstract

With the advent of lightweight cryptography in edge devices, attackers can hide malicious code under encrypted network communications to perform malware attacks. This makes IoT botnet attacks extremely challenging to detect by means of traditional signature-based techniques. In this paper, we propose a novel IoT botnet detection framework that uses vision transformers to detect malicious communications captured in encrypted network flow images. Our approach achieved ∼98% accuracy and around 94% reduced inference latency compared to state-of-the-art approaches. Further, we have validated the practicality of our approach by testing it on Jetson Orin Nano acting as an edge gateway and achieved reduced inference latency of 25.16 ms and area overhead of 88.13 MB.

## 1 Introduction

With the advancement in embedded technologies and the improved availability of Internet connectivity, there has been a rapid integration of smaller and smarter IoT devices in several domains of daily life, such as smart home appliances, smart cities, and logistics. According to IoT Analytics (Christian, 2025), the number of connected IoT devices reported an increase of approximately 13% year-over-year, exceeding 18 billion in 2024. However, the resource-constrained nature and heterogeneous architecture of these devices and the huge competition to release new IoT products to the market limit the adoption of unified security solutions for IoT (Dange & Chatterjee, 2019). This results in many security flaws, such as open ports and default credentials that increases vulnerability for malware attacks. Among the various types of IoT malware attacks, botnets are the most serious and large-scale attack that can even bring down a secure and critical infrastructure. For instance, in October 2016 Mirai caused the biggest Distributed Denial of Service (DDoS) attack against the Dyn service provider which disabled the websites including Twitter, Netflix, and GitHub for several hours (Dange & Chatterjee, 2019).

Botnets are a collection of compromised machines known as bots that run malicious code under the command and control of a botmaster. When the compromised machines are IoT devices, it becomes an IoT botnet. The botnet lifecycle consists of 3 phases; scanning, propagation, and attack. In the scanning phase, botmaster scans for vulnerable IoT devices and, once found, compromises it (Negera et al., 2022). The propagation phase proceeds by installing malware on these systems, and this chain continues by the new bots finding new victim bots. In the attack phase, botmaster instructs the bots through a command and control server to initiate attacks such as DDoS, spam, and cryptocurrency mining, etc. According to Zscaler ThreatLabz Enterprise IoT and OT Threat Report (Blog, 2023), there is a 400% surge in IoT malware attacks in past few years, with botnets such as Mirai and Gafgyt dominating the attack space, as indicated in Figure 1. Evidently, there is an urgent need for IoT botnet detection solutions that are quick with low computation overheads suitable for IoT/edge devices.

Furthermore, IoTNOW (Nelson, 2020) reported that 98% of IoT traffic was unencrypted in 2020 due to its resource-constrained nature, transforming IoT into Internet of Threats, and many leading companies were planning to improve the security posture of IoT devices by means of encryption. With lightweight cryptography gaining popularity (R, 2022), there has been a significant interest among the researchers (Abutaha et al., 2022; Suzaki et al., 2012), and industries (AVNET, 2023) in the area of developing lightweight

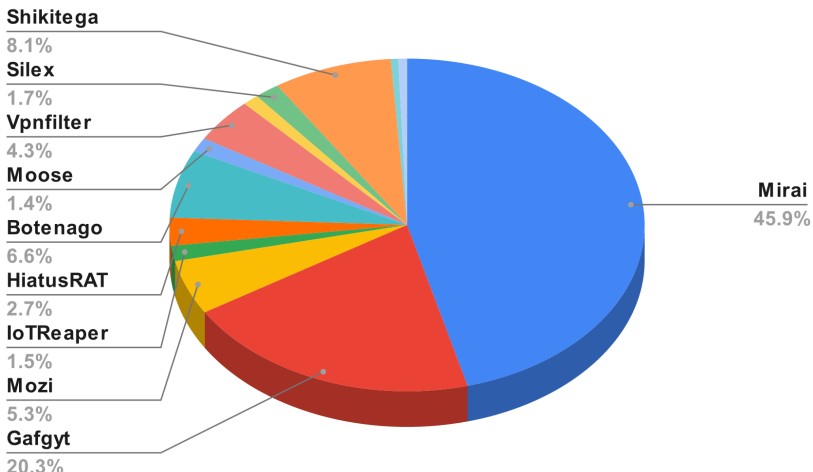

Figure 1: Top IoT malware families (Blog, 2023).

encryption algorithms for IoT networks. Encryption in communication can play a critical role as it may allow malicious actors to hide malware codes in the payload which may force the existing botnet detection techniques to fail.

State-of-the-art botnet detection techniques majorly deploy network based monitoring system to analyze the traffic for abnormalities by either matching against a signature database or using machine learning techniques. These approaches can be further categorized into two broad groups: Network Flow analysis and Deep Packet Inspection (DPI) (Koroniotis et al., 2019a). Botnets typically generate large amounts of network traffic in bursts in all phases of its lifecycle and can thus be modeled as a series of network flows. Here, the network flow group together different packets that share Source IP and Port, Destination IP and Port, and Protocol and collect metrics such as bytes per packet, flows per hour, flows per address, etc. (Dange & Chatterjee, 2019). The benefit of this approach is that privacy is not a concern, as the actual payload is not investigated and works well on encrypted communication if the header is unencrypted (Koroniotis et al., 2019a). The disadvantage is that the information in the payload will be lost. DPI investigates the content of each packet, which yields more complete results. However, it adds considerable overhead, breaches the privacy of users, and fails in case of encrypted communications and zero-day attacks (Koroniotis et al., 2019a). Most of the works available in the literature utilize either of these two approaches for IoT botnet detection. However, to the best of our knowledge, there is no work that analyzes both the header and payload in a timely manner without compromising on the accuracy. Additionally, when it comes to resource-constrained systems, inference time plays an important factor. Thus, an accurate approach with reduced memory and time utilization is of critical importance.

Therefore, we propose a novel and efficient IoT botnet detection approach that works well in detecting encrypted botnet communications with minimal computational overhead and latency. Our framework uses RGB image representations of network flow PCAP files as the input to predict the botnet attack category, and hence does not breach the privacy of users. In this work, we are focusing on the attack phase of the botnet life cycle, which is the most destructive by inspecting the network traffic emanating from the IoT bots.

## 1.1 Contributions

Our novel contributions include

- Designed a novel IoT botnet detection framework ViT-EBoT which examines the bidirectional network flows/sessions by converting them to images and processing them using advanced deep learning techniques.

- ViT-EBoT leverages vision transformers to detect whether the network communication belongs to 11 classes such as Benign, DoS, DDoS, Scan, and Data Exfiltration attacks using the BoT-IoT dataset.

- Investigates the entire packet, i.e., header and payload to ensure that analysis is complete and no information is lost.

- To the best of our knowledge, this is the first work for IoT botnet detection that works well on encrypted IoT communications.

- Achieved an accuracy of ∼98% for both unencrypted and encrypted communications with 94% reduced latency compared to state-of-the-art approaches.

- Optimized the ViT-EBoT model using TensorRT, implemented it on the Jetson Orin Nano to simulate practical deployment, and achieved 98% reduction in latency.

## 2    Related Work

With the number of botnet attacks increasing at an alarming rate, researchers and industry experts are designing novel and efficient techniques for detecting them. These techniques can be broadly classified into host-based and network-based (Dange & Chatterjee, 2019). Host-based solutions generally suffer from high computational requirements and scalability challenges. For instance, Nguyen et al. (2020) proposed a solution that extracts high-level features from function call graphs, called Printable String Information (PSI) for each executable file to detect botnets. Although it achieves decent accuracy of 98.7%, it fails to scale with increasing number of bots.

Network-based approaches are more preferred considering the resource-constrained nature and volume of the IoT devices (Dange & Chatterjee, 2019). Moreover, botnets are not an individual attack, but a consolidated attack by different devices. Most of the works belonging to the network-based category use popular IoT botnet datasets such as BoT-IoT (Koroniotis et al., 2019b) and N-BaIoT (Meidan et al., 2018) to extract network flow features and apply different dimensionality reduction, feature selection, and dataset balancing techniques along with machine and deep learning models for prediction. For example, Popoola et al. (2020) utilized a long-short-term memory autoencoder (LAE) for dimensionality reduction and deep bidirectional long short-term memory (BLSTM) for classification. Alshamkhany et al. (2020) applied Principal Component Analysis (PCA) for dimensionality reduction and tried different machine learning models with decision tree, attaining the better accuracy of 99.89%.

All the above mentioned network-based works used the extracted features in the dataset and did not consider the actual complexities involved in extracting those features. However, the work by Hasan et al. (2022) employ the features from the *conn.log* of Zeek network analysis framework and proposed a botnet detection framework by modeling the bot network connections as a Markov Chain and applied Class-specific Cost Regulation ELM (CCRELM) for detection. The framework achieved 97.7% accuracy. However, these works focused on improving the accuracy of the model and did not consider the real-time performance of their approaches in terms of throughput and resource utilization that is imperative when it comes to IoT devices. For instance, tools like Zeek require at least 64GB memory (Documentation) to perform that is not possible on a resource-constrained system. Further, these techniques fail to extract any information from the payload, and hence are less robust. On the other hand, DPI approaches mostly require a signature database against which it will match the payload signatures and hence fail in the case of encrypted and zero-day attacks (Koroniotis et al., 2019a).

All of the above works assume that the IoT communications are unencrypted. However, with the lightweight cryptography gaining momentum in the recent years, works which statically analyze the encrypted IoT network traffic for IoT device identification such as the work by Pinheiro et al. (2019) and network traffic classification into mail, chat, video streaming, etc. (Wang et al., 2017; Ma et al., 2021) are on the rise. Thus, we need an approach to detect IoT botnet attacks that applies a resource efficient feature extraction phase and at the same time achieving better performance on encrypted communications as well.

# 3 Proposed Methodology

In this section, we discuss our novel IoT botnet detection framework in detail. Our proposed framework consists of two phases, as shown in Figure 2:

1. Feature Extraction phase

2. Detection phase

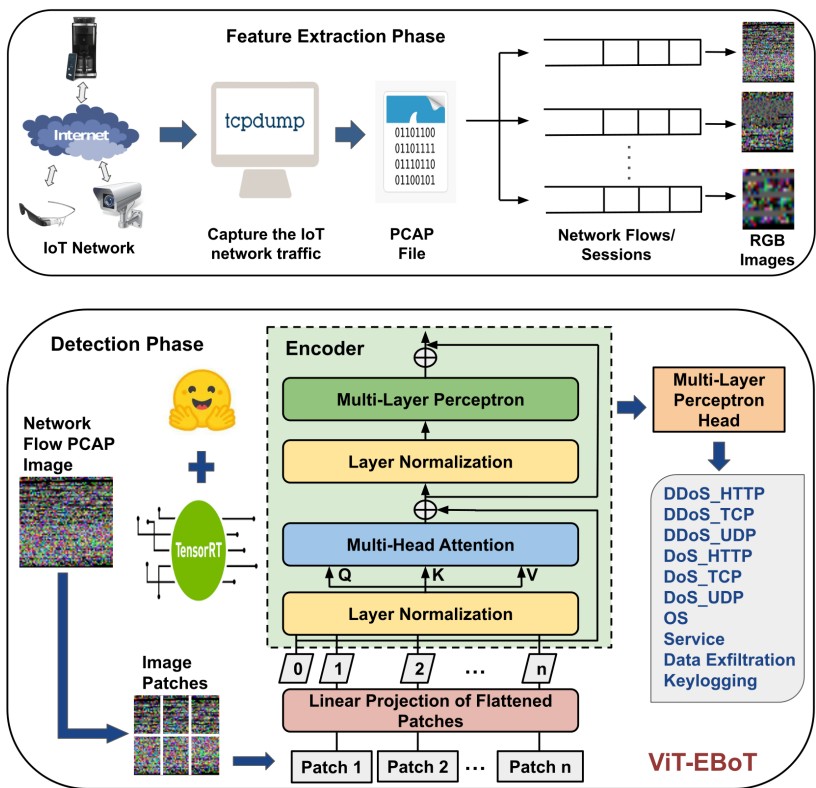

Figure 2: Proposed Architecture ViT-EBoT

The feature extraction phase preprocesses the input to make it compatible with the ViT-EBoT botnet detection model. The detection phase will utilize these inputs to predict the botnet attack category. Since we are focusing on a network-based framework, the input required for our system is the network traffic. Therefore, in the feature extraction phase, we capture the IoT network traffic in the form of Packet Capture (PCAP) files using *tcpdump*, a data-network packet analyzer. This PCAP file will contain the network packets that flow between different source-destination pairs. However, we are interested in bidirectional network flows. We have used *SplitCap* (NETRESEC, 2010) tool to split the captured PCAP file into several small PCAP files corresponding to each session. In contrast to state-of-the-art botnet detection approaches that analyze flow summary statistics or packet payload, we convert each extracted network flow PCAP file to an RGB image with square dimensions following the procedure shown in Figure 3. Here, we encode each 24 bits (8 bits per channel) of the PCAP file as one pixel. The RGB image comprises both the packet header and payload information, making the analysis more complete and robust. Figure 4 shows that clearly distinguishable patterns can be extracted from these images that correspond to each class. At the same time, we are not actually parsing the packet payload and extracting signatures from it, thereby safeguarding the privacy of users. However, DPI fails terribly when faced with encrypted communications due to the unavailability of signatures in clear text (Koroniotis et al., 2019a). The experimental results in Section 4 corroborates that our approach works well in the case of encrypted communications.

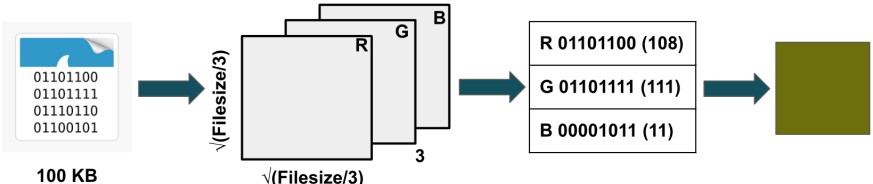

Figure 3: Network Flow PCAP to RGB image conversion

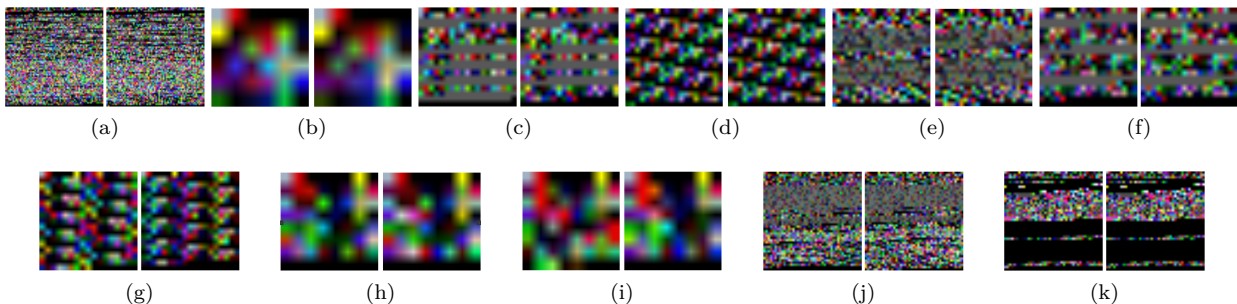

(a)          (b)          (c)          (d)          (e)          (f)

(g)          (h)          (i)          (j)          (k)

Figure 4: Patterns observed in network flow PCAP images belonging to different classes. (a) Benign. (b) DDoS_HTTP. (c) DDoS_TCP. (d) DDoS_UDP. (e) DoS_HTTP. (f) DoS_TCP. (g) DoS_UDP. (h) OS. (i) Service. (j) Data Exfiltration. (k) Keylogging.

In the detection phase, these RGB images will be fed to ViT-EBoT, a vision transformer-based model for predicting the IoT botnet attack category. For our approach, we have used the ViT model pre-trained on ImageNet-21k at resolution 224x224 and a patch size of 16x16 (Hugging-Face) and fine-tuned the pretrained model on our BoT-IoT (Koroniotis et al., 2019b) dataset. Dataset details are given in Section 3.1. The input images will be resized to 224x224 and normalized across the channels with a mean and standard deviation of [0.5, 0.5, 0.5]. A learnable class token is prepended to the embedded patches to capture the entire image representation. ViT consists of alternating multi-head attention and multi-layer perceptron layers with a layer normalization before and skip connection after it (Alexey, 2020). Finally, ViT-EBoT performs multiclass classification using the learned class token and predicts the botnet attack category.

Next, we will see how our proposed framework ensures privacy. Encrypted network communications protect the privacy of communicating parties. Approaches like deep packet inspection need to analyze the payload, and hence need to find the encryption key and decrypt the encrypted traffic. This will breach the privacy of users. However, our approach can work with encrypted traffic for botnet attack detection and does not need to decrypt it thereby safeguarding user privacy. We assume that even if the payload is encrypted, botnet communications share some patterns which can be identified by our ViT-EBoT framework as shown in Figure 5. It is true that since our approach reencodes the data from linear to graphical format, the obtained images can be directly converted back to the PCAP format. However, whatever we retrieve will be encrypted traffic, and we are not going to decrypt it. On the other hand, if the network traffic is unencrypted, already data is transmitting in clear text, and hence there are no privacy concerns. In the next sections, we will discuss the dataset and encryption methodology used in our work.

## 3.1 Dataset

We have used the BoT-IoT dataset, a popular dataset generated by the Cyber Range Lab of UNSW Canberra with more than 72,000,000 records of normal and botnet traffic and 69.3 GB in size. This dataset provided the original PCAP files corresponding to the botnet attack phase, which can be used to extract network flows and create images. For ease of handling and to reduce the training time of our model, we have extracted only 10% network flows from the dataset (6.6 GB). The dataset includes the 11 classes such as Benign, DDoS_HTTP (DDoS attack using the HTTP protocol), DDoS_TCP, DDoS_UDP, DoS_HTTP, DoS_TCP, DoS_UDP,

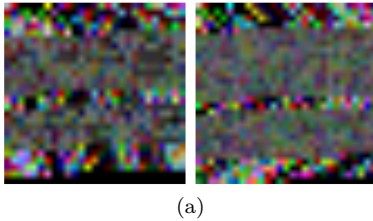 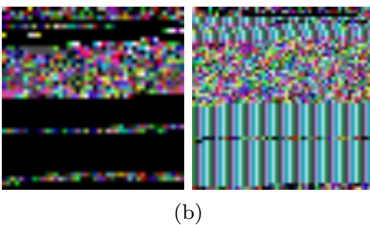

(a)                                                    (b)

Figure 5: Original and encrypted network flow images. (a) DoS_HTTP (b) Keylogging

Scan, Service, Data Exfiltration, and Keylogging. All the classes, except Data Exfiltration and Keylogging, consist of 1500 images. Data Exfiltration and Keylogging classes have 239 and 1638 images, respectively.

### 3.2   Encryption Procedure

We have extracted 15,377 network flow images corresponding to 11 classes from the BoT-IoT dataset. However, there are no datasets available in the literature that consists of encrypted IoT botnet communications. Thus, we are the first to design and implement an IoT botnet detection framework on a resource-constrained setup which can even detect botnets hiding under encrypted communications with better accuracy. Hence, we extended our dataset with synthetic encrypted PCAP files. We have chosen Advanced Encryption Standard (AES), the most popular symmetric encryption algorithm, and a lightweight encryption algorithm TWINE (Suzaki et al., 2012) for encrypting the TCP/UDP payload of the network flow PCAP files. The pseudocode for the encryption procedure is shown in Algorithm 1.

---

**Algorithm 1** Encryption Procedure

---

1:  **procedure** ENCRYPTION
2:      *NetFlow_PCAP* ← Read PCAP file using Scapy
                        *rdpcap()*
3:      *New_PCAP* ← Empty
4:      **for** *curr_packet* in PCAP file **do**
5:          **if** *TCP/UDP* packet and nonempty payload **then**
6:              *payload* ← Extract TCP/UDP payload
7:              *encrypted_payload* ← *AES/TWINE (payload)*
8:              Delete *packet length* and *checksum*
9:              *New_PCAP* ← Rebuild packets
10:         **else**
11:             *New_PCAP* ← *curr_packet*
12:         **end if**
13:     **end for**
14: **end procedure**

---

Figure 6 shows the original and encrypted versions of a PCAP file. Here, the payload "Hello" in Figure 6a is encrypted in Figure 6b, and the payload length has increased to 16 bytes. This is the reason for recalculating the payload length and checksum of the packet. Finally, we randomly encrypted the original 15,377 samples using AES or TWINE and created 15,377 encrypted samples. Here, we are encrypting only the packet payload, and thus our trained neural network depends on both unencrypted packet headers and unencrypted/encrypted payload for botnet detection.

## 4   Experiments and Results

In this section, we will first compare the performance of our approach against state-of-the-art techniques such as flow header analysis and deep packet inspection. Next, we have explored the performance of different

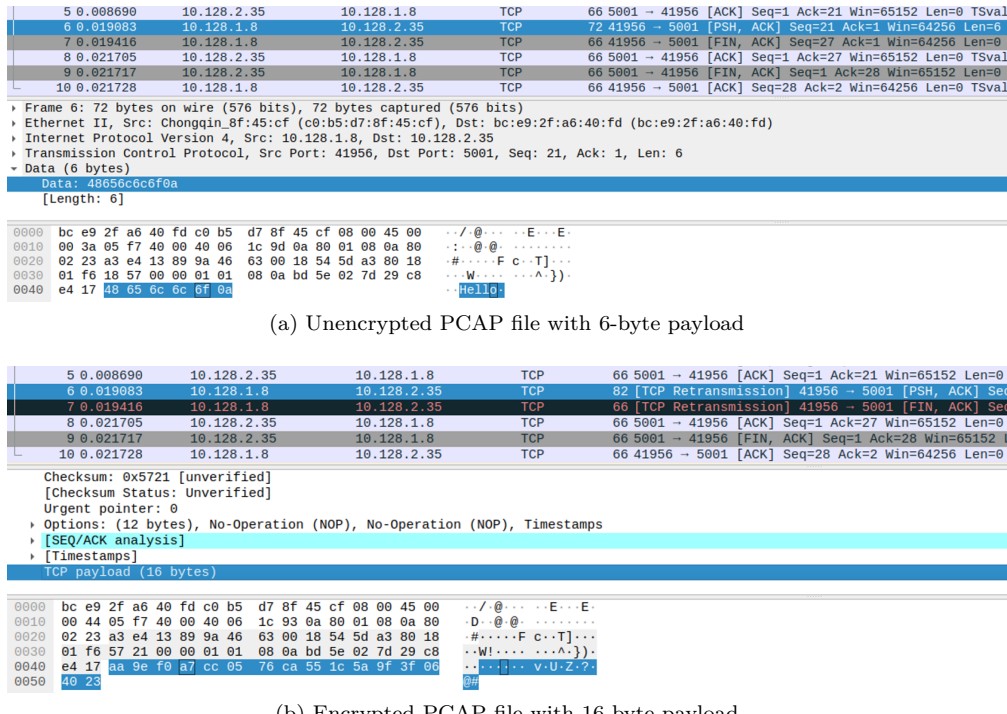

(a) Unencrypted PCAP file with 6-byte payload

(b) Encrypted PCAP file with 16-byte payload

Figure 6: Original and encrypted versions of a PCAP file.

deep learning models for encrypted botnet detection in terms of accuracy to emphasize our choice of vision transformer-based model. Then, we will analyze whether all the packets in the network flow are actually necessary for botnet detection. Finally, we will discuss the implementation details on the Jetson Orin Nano device and visualize the accuracy, inference latency, and throughput metrics.

Our dataset consists of 15,377 samples from the BoT-IoT dataset and their synthetically encrypted variants. We have used the *"google/vit-base-patch16-224-in21k"* Hugging Face pretrained model and fine-tuned it on our 30K dataset for 10 epochs. We have trained our models on NVIDIA RTX A4500 GPU with $12^{th}$ Gen Intel® Core™ i9-12900K × 24 processor. The inference part was deployed on a Jetson Orin Nano.

### 4.1 Comparison with Flow Header Analysis and Deep Packet Inspection

We have compared our network flow PCAP image-based approach using vision transformers with two state-of-the-art approaches such as flow headers and deep packet inspection (Koroniotis et al., 2019a). We have used Python *CICFlowMeter*, a network traffic flow generator and analyzer to extract more than 80 statistical network traffic features such as duration, number of packets, number of bytes, length of packets, etc. Then, we have trained several machine learning models such as Random Forest, Decision Tree, and Naive Bayes on these features for botnet attack detection. Next, we have used nnDPI (Bahaa et al., 2020), a deep packet inspection module using word embedding, convolutional, and recurrent neural networks. Results for unencrypted and encrypted flows are shown in Figure 7a and 7b respectively. Table 1 summarizes the overall F1-score and average feature extraction time for the state-of-the-art approaches.

Here, the PCAP image-based approach achieved marginally better F1-score of 98.39% compared to flow header analysis. However, the feature extraction time of just 61.71 ms stands out for our approach, which is extremely useful for resource-constrained edge devices. Note that the feature extraction time in Table 1 includes the pre-processing time required to convert the network flows to the expected input format of the state-of-the-art models alone and does not include the prediction time that is almost similar for all. Splitting network traffic into flows takes some time depending on the size of the captured network traffic and the number of flows in it. For example, it takes 2.34 seconds to split a PCAP file of size 243.7 MB

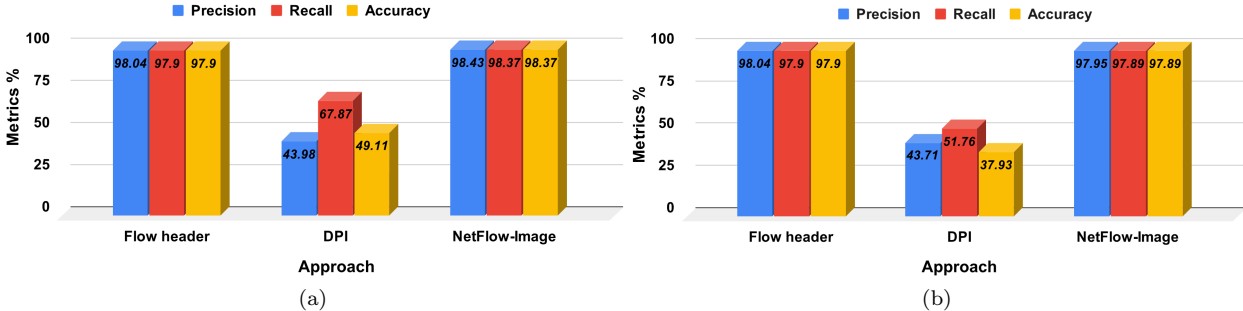

Figure 7: Performance of the state-of-the-art models on (a) Unencrypted and (b) Encrypted network traffic.

Table 1: Comparison with state-of-the-art models in terms of F1-score and feature extraction time

| Approach | F1-score | | Feature Extraction |
|---|---|---|---|
| | Unencrypted | Encrypted | Time (ms) |
| Flow header | 97.9 | 97.9 | 471.35 |
| DPI | 42.4 | 35.58 | 46.76 |
| NetFlow-Image | 98.39 | 97.9 | 61.71 |

comprising 10,087 flows on average. Network flow header-based approach maintained the same F1-score over encrypted samples due to the fact that we have only encrypted the payload and the header information remains untouched. The reduced feature extraction time of nnDPI is due to the fact that it considers only the first 1500 bytes of each packet for analysis. This per-packet analysis is not feasible on a resource-constrained setup, and 1500 bytes are not sufficient to predict whether a botnet attack is in progress because it is a collective effort by different infected bots. This resulted in an extremely low F1-score for nnDPI. Thus, our approach proves to be faster with almost 87% reduced latency compared to the flow header-based approach. Hence, it works efficiently when faced with large network traffic volume on resource-constrained systems.

## 4.2  Performance comparison of deep learning models on Network Flow PCAP images

Next, we will discuss the performance of different computer vision models in the IoT botnet detection area. For this, we have used 6 pretrained models from Hugging Face on ImageNet dataset and fine-tuned it on our three datasets which are:

(1) 15377 original network flow PCAP images

(2) images corresponding to the encrypted versions of 15377 original network flow PCAPs in dataset (1).

(3) Combination of dataset (1) and (2) comprising original and encrypted PCAP images, respectively, totaling 30,754 samples.

Figure 8 shows the training and validation F1-score for the 3 datasets. Table 2 shows the final results of the 6 models on a test dataset with 6151 images for the third dataset. ViTs work exceptionally well with an F1-score of 97.87% and a testing loss of 0.09 due to the attention layers that capture global features of the input image.

Since the network flows collected correspond to 11 attack classes, we have further examined whether one class is detected better than the other. The per-class accuracy for the 11 classes of our model on a test dataset with 6151 images is shown in Figure 9a. Here, we could see that our framework worked equally well on all the 11 classes with a per-class accuracy of ∼99%. Confusion matrix shown in Figure 9b corroborates this finding.

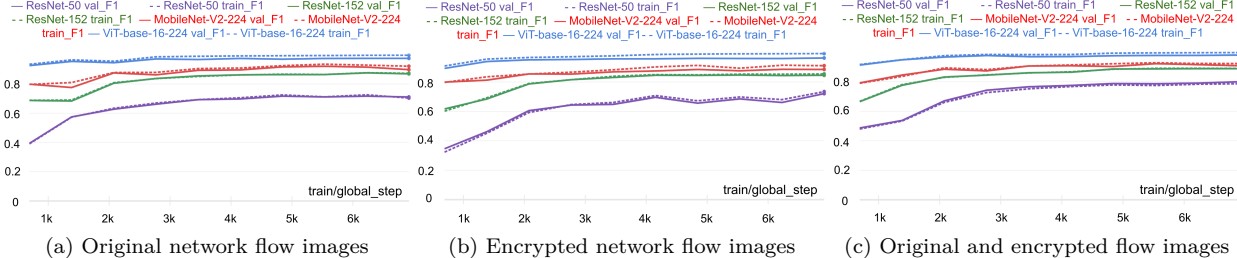

(a) Original network flow images    (b) Encrypted network flow images    (c) Original and encrypted flow images

Figure 8: Training and validation F1 score of the four pretrained models on 3 datasets

Table 2: Results of deep learning models on the test dataset (3) with 6151 images

| Model | Accuracy | Loss | F1-score | Precision | Recall |
|---|---|---|---|---|---|
| Vit-base-patch16-224 | 97.87 | 0.09 | 97.87 | 97.89 | 97.87 |
| MobileViT | 90.42 | 0.29 | 90.32 | 90.74 | 90.42 |
| deit-tiny-patch16-224 | 97.25 | 0.11 | 97.24 | 97.26 | 97.25 |
| Mobilenet-v2 | 91.69 | 0.26 | 91.74 | 91.93 | 91.69 |
| Resnet-152 | 88.03 | 0.34 | 87.78 | 88.11 | 88.03 |
| Resnet-50 | 79.16 | 0.6 | 78.64 | 80.17 | 79.16 |

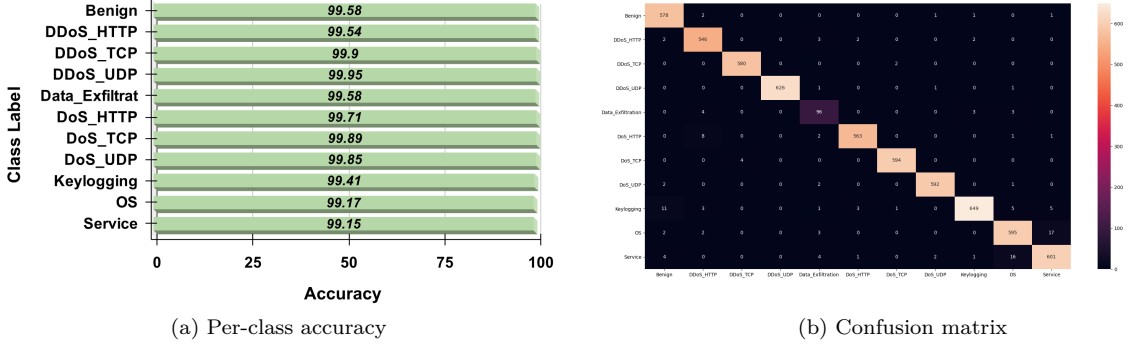

(a) Per-class accuracy    (b) Confusion matrix

Figure 9: Per-class accuracy and confusion matrix of the ViT-EBoT on the test dataset (3) with 6151 images

### 4.3 Analysis of flow length distribution

From Table 1, we could see that generating the network flow image requires 61.71 ms on average which may vary depending on the flow length. Flow length is the total number of packets in a network flow, and for our dataset, it is varying significantly from 1 to 363141. Therefore, we have analyzed the flow length distribution to select the optimal number of packets to consider for botnet detection. From Figure 10, we can notice that most of the flows (∼90%) have less than 50 packets. Thus, we performed experiments to see the effect of the flow length on the prediction accuracy. We first filtered the first 50 packets and reduced the number of packets at each step to see how the accuracy of ViT-EBoT and the feature extraction time are impacted. Results are shown in Table 3.

Thus, we can see that even with the first 5 packets in a flow, we achieved 98.52% accuracy and 0.069 loss. From Table 1, we could see that the difference between the flow header and the proposed approach was mainly the time overhead. To maintain a fair comparison, we performed similar experiments to see how the time overhead and accuracy are impacted for the flow header approach on fewer packets per PCAP file. Results are shown in Table 3.

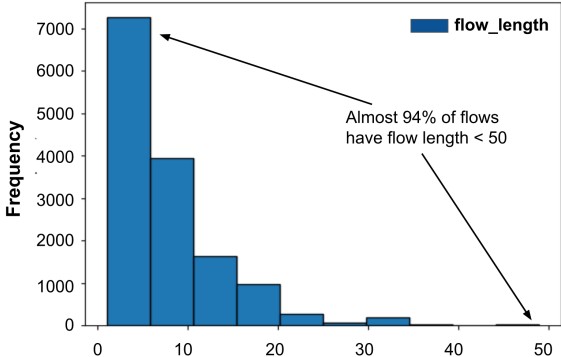

Figure 10: Flow length distribution of the original dataset

Table 3: Effect of flow length on the performance of ViT-EBoT and flow header-based approach

| Number of packets | ViT-EBoT | | Flow Header approach | |
| --- | --- | --- | --- | --- |
| | F1-score | Feature Extraction Time (ms) | F1-score | Feature Extraction Time (ms) |
| 5 | 98.52 | 0.22 | 97.97 | 370.46 |
| 10 | 97.91 | 0.28 | 97.49 | 369.97 |
| 20 | 97.89 | 0.33 | 97.6 | 370.26 |
| 30 | 97.81 | 0.37 | 97.61 | 372.93 |
| 40 | 97.85 | 0.40 | 97.7 | 372.73 |
| 50 | 98.02 | 0.44 | 97.75 | 371.37 |
| ALL | 97.75 | 61.71 | 97.86 | 471.35 |

From Table 3, we could see that the feature extraction time manifested a significant reduction of 99.64% while reducing the number of packets for the network flow image-based approach. However, for the flow header approach, the reduction is only 21.4%. This is because even with just five packets per network flow PCAP file, computations required to summarize the flow statistics remain almost the same. Moreover, there is not much impact on the accuracy in both scenarios. In short, the conversion time for network flow to image reduced significantly upon extracting only the first few packets, and thus we can detect a botnet attack effectively by analyzing only these packets in most scenarios at a faster pace. The first five packets are sufficient since most PCAPs will contain both the header information, i.e., TCP handshake in the case of TCP as the transport protocol, and some initial payload in these first packets. In contrast, the work by Wang et al. (2017) and Ma et al. (2021) fixes the number of bytes extracted from each network flow or session PCAP file. The former uses the first 784 bytes of each flow or session, while the latter extracts the first 500 bytes of the payload as key data. However, here we are completely analyzing the first 5 packets.

## 5 Implementation

The following subsections will discuss how our framework will be deployed in practical scenarios, optimizations applied to fit the model on Jetson Orin Nano, and finally the results achieved.

### 5.1 Practical Deployment

IoT devices are increasingly becoming compact and energy efficient with high performance owing to the widespread adoption of SoCs in IoT development. SoCs provide better performance at an affordable price and are integrated with embedded GPUs/NPUs for accelerating AI applications. We have optimized and implemented our ViT-EBoT framework on Jetson Orin Nano that is built around a low-power version of the NVIDIA Orin SoC with an embedded GPU. It is the most computationally powerful SoC with 8 GB 128-bit LPDDR5 RAM and NVIDIA Ampere architecture with 1024 CUDA cores and 32 tensor cores (NVIDIA,

2023a). For testing the practical feasibility of our framework, we have performed our inference on Jetson Orin Nano to ensure that, in practical scenarios, we can deploy this model on the other SoCs available in resource-constrained IoT devices such as Qualcomm SoCs in IP cameras for large-scale integration. Further, we have reduced the inference latency on the board by using TensorRT, which includes a deep learning inference optimizer and runtime that delivers low latency and high throughput for inference applications on NVIDIA GPUs (NVIDIA, 2023c). Figure 11 illustrates the scenario of practical deployment of ViT-EBoT.

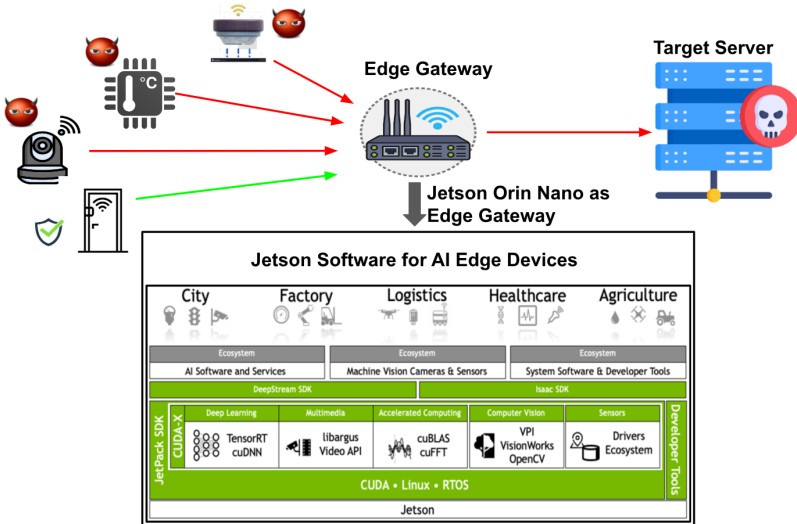

Figure 11: Practical deployment scenario (NVIDIA, 2023b): ViT-EBoT deployed on Jetson Orin Nano acting as edge gateway

Here, we assume some of the IoT nodes connected to the edge gateway to be bots. Jetson Orin Nano acts as an edge gateway and monitors all the traffic passing through it to identify if a device connected to it has become a zombie/bot and performing a botnet attack on a critical server in the internet. In practical scenarios, our ViT-EBoT framework will be deployed on the edge gateways that will inspect the network traffic targeted for the IoT nodes connected to that gateway. Once the gateway captures the network traffic using *tcpdump* utility, *Splitcap* utility will split each PCAP into multiple network flow sessions and finally convert these PCAP files to network flow images for inference purposes on the gateway, as shown in the feature extraction phase of Figure 2. This flow image analysis will act as a defense mechanism and block any botnet traffic. For simulation purposes, we have stored the network PCAP files from BoT-IoT test dataset on the Jetson board. Then, we have split each PCAP into multiple network flows, converted to RGB images, and finally performed inference on these test images.

## 5.2 Optimization workflow on Jetson Orin Nano

The overall implementation workflow is shown in Figure 12. First, we will load the trained ViT-EBoT and feature extractor that will preprocess the images for inference task on the Jetson board. Next, we will export the 32 bit floating point model to a format that TensorRT can use, such as Open Neural Network Exchange (ONNX). Then, we used TensorRT's *trtexec* tool to parse the ONNX model and generate an optimized TensorRT engine file which can be used to perform the final inference on the Jetson board. We have chosen a batch size of 64 and an input shape of (3, 224, 224).

In order to optimize the model to improve performance and reduce latency, we have applied Post Training Quantization (PTQ) using *pytorch-quantization* (NVIDIA, 2021) toolkit for our framework. PTQ is applied over a trained full-precision model. Per channel quantization is applied to the weight tensors using the weight distribution of the trained model. Thus, as shown in Figure 12, we have separate scale $S$ and zero point $Z$ for each channel. For calculating the interlayer activation distributions, we have applied *histogram* calibration using a small calibration dataset (Neta Zmora & Rodge, 2021). INT8 quantization is applied to

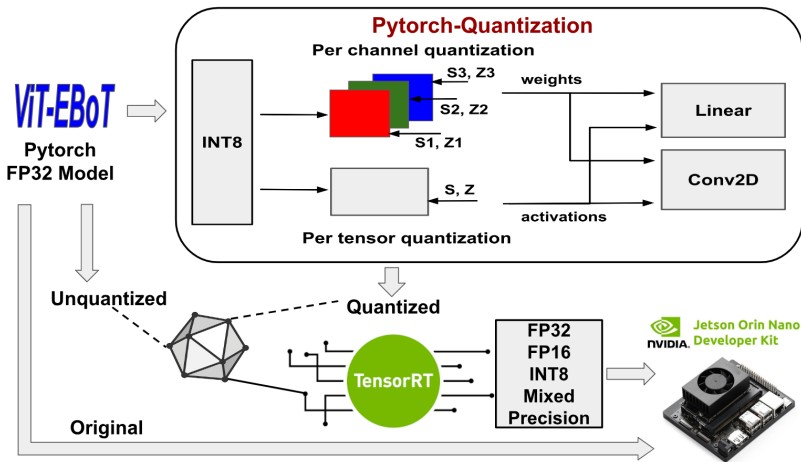

Figure 12: Optimization workflow of ViT-EBoT on Jetson

the weights and activations of the linear and 2D convolution layers in Figure 13a to better utilize the integer tensor cores of NVIDIA as shown by the QuantizeLinear/DequantizeLinear ONNX operations in Figure 13b. Visualization of the neural network is provided by the Netron (Roeder, 2020) tool. The TensorRT will import the fake quantized model exported to ONNX and execute it for inference in an optimized manner.

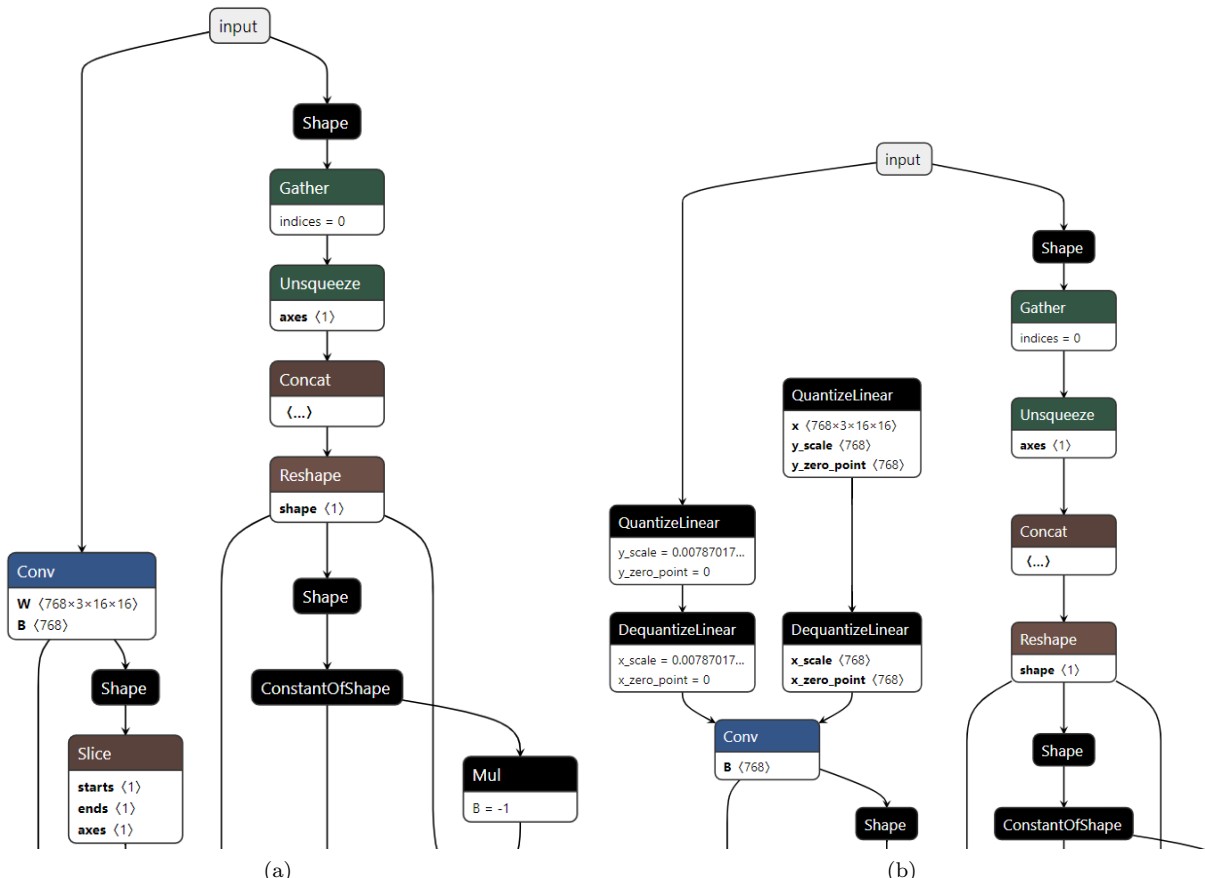

Figure 13: Original and quantized network visualizations (a) Original network (b) Quantized network

### 5.3 Results

Figure 14-15 demonstrates the performance metrics of the model on the Jetson board for 20 batches with batch size 64. First, we have the unoptimized model running on NVIDIA GPU. Then, we have optimized the 32 bit floating point (FP32) model with TensorRT. Next, we have converted the FP32 model to 8 bit integer (INT8) representation. Finally, the linear and convolution layers in the FP32 model are fake quantized to INT8 representation, and the rest of the layers were maintained at FP32 precision. From Figure 14a, we could see that the quantized INT8 model achieved 98.74%, i.e., only 0.47% less accuracy than the original unoptimized model. It also gained 3.9x savings in model size, as can be seen from Figure 14b. Furthermore, with the unoptimized model, the inference latency is 1.69s and a throughput of 0.59 frames per second (FPS). However, the INT8 quantized model reduced the latency by 98.51% and achieved 41.34 FPS as shown in Figure 15. Thus, we were able to achieve a fast and accurate model for botnet detection with reduced model size of 88 MB. Thus, ViT-EBoT works efficiently in a resource-constrained setup simulated on a Jetson Orin Nano board.

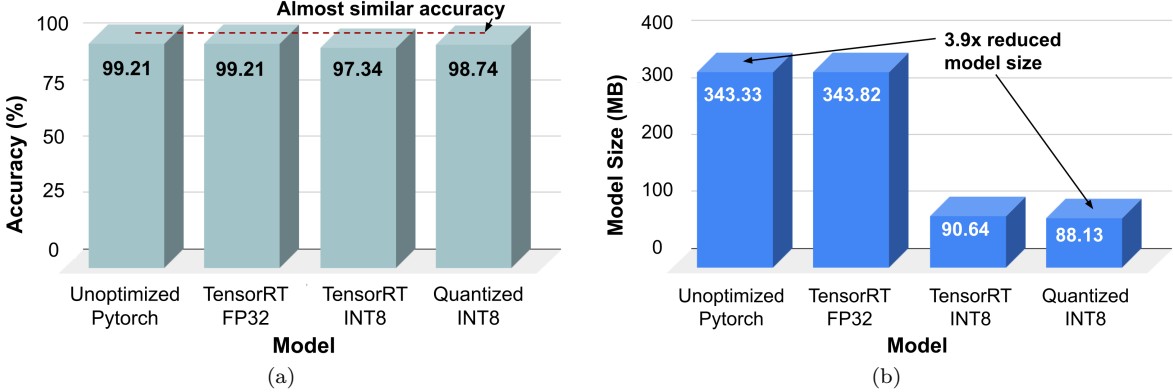

Figure 14: Accuracy and model size comparison on Jetson Orin Nano (a) Accuracy (b) Model Size

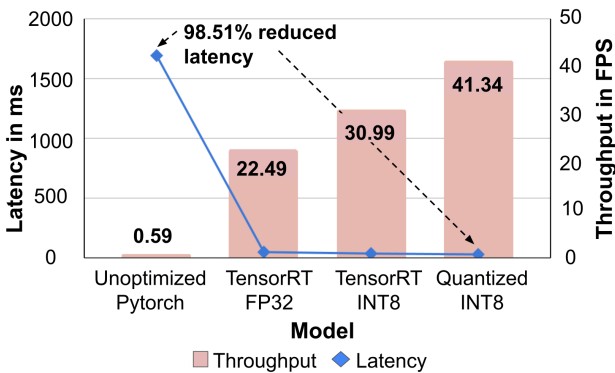

Figure 15: Effect on the latency and throughput of the models

Next, we have analyzed the cost of deploying both ViT-EBoT and Flow header approaches that achieved similar accuracy. Clearly, the cost of deploying machine learning models used in the flow header approach will be less than our ViT-EBoT framework in terms of area. However, feature extraction time for the flow header-based approach is almost 8 times greater than our approach. To confirm this, we have used *CICFlowMeter* on the board to capture network flow statistics and it took almost 76 minutes for extracting these features from a 1.1 GB PCAP file with 42,490 network flows i.e., 107.01 ms per network flow. Hence, feature extraction phase of flow header-based approach is not memory effective on an edge system. Lower feature extraction time with better accuracy ensures botnet detection at a fast pace. Moreover, we do not

need to deploy the ViT-EBoT model on each IoT device. Rather, it is deployed on each edge gateway device that forwards the network traffic to individual IoT nodes, thereby reducing the cost.

## 6 Conclusion

In this paper, we have developed an IoT botnet detection framework ViT-EBoT that converts the bidirectional network flows to RGB images and applies vision transformers to predict the botnet attack category. Our framework achieved around 98% accuracy for both encrypted and unencrypted network samples with 94% reduced inference latency compared to state-of-the-art approaches. In addition, we have deployed the framework on a Jetson Orin Nano board to validate its feasibility as a practical edge gateway and optimized it through TensorRT INT8 quantization. The optimized model achieved an accuracy of 98.74% over 1280 images with a reduced latency of 25.16 ms i.e., 98% reduced latency compared to the unoptimized version and an area overhead of 88.13 MB.

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
