# OpenReview forum: "ViT-EBoT: Vision Transformer for Encrypted Botnet Detection in Resource-Constrained Edge Devices"
_TMLR — Withdrawn by Authors_

### Review · Reviewer_2WsS · 2025-03-23

**Summary Of Contributions:**

The paper proposes using vision transformers to detect botnets from the encrypted network packet data. The main idea is to convert
 the network data into RGB images. The authors then use a vision transformer pre-trained on ImageNet-21k and fine-tuned on the BoT-IoT dataset. The proposed method has a much faster feature extraction speed, and so it is suitable to be deployed on resource-constrained edge devices.

**Audience:**

No

**Broader Impact Concerns:**

There are no concerns about the ethical implications of this work.

**Claims And Evidence:**

Yes

**Requested Changes:**

See the Weakness section above.

**Strengths And Weaknesses:**

Strength:
- The paper is clearly written and easy to follow.
- The proposed method achieves a high F1-score with a fast feature extraction time

Weakness:
- There does not seem to be too much insight and contribution to the area of machine learning. The paper should be submitted to IoT or network security conferences/journals.
- The paper only considers the transformer architecture, while for images, CNN is another mainstream architecture and should be considered too. Otherwise, the authors should explain the motivation for using this particular architecture more clearly.
- There are many prior works on using deep learning models on raw sequential network data, such as [1]. The authors should compare those methods to show that the proposed conversion to RGB is superior.
- The paper cites many online articles where the correctness is not verified.

 [1] Shahhosseini et al., A Deep Learning Approach for Botnet Detection Using Raw Network Traffic Data. Journal of Network and Systems Management. 2022.

---

### Review · Reviewer_sL92 · 2025-03-23

**Summary Of Contributions:**

This paper presents a new IoT botnet detection approach ViT-EBoT that detects the encrypted botnet communication with small computational overhead and latency. It uses the image version of PCAP files and examines the bidirectional network flows and sessions using deep learning techniques. The detection accuracy was similar or higher compared with state-of-the-art detectors, even with reduced resources. They experimented class classification using 11 different attack class datasets, which show about 90% accuracy with various models. Furthermore, implemented their framework on Jeston Orin Nano to experiment with practical implementation in limited resources, and showed high accuracy with low latency.

**Audience:**

Yes

**Broader Impact Concerns:**

No.

**Claims And Evidence:**

Yes

**Requested Changes:**

1. The model is experimented with only using a portion of one dataset, BoT-IoT. It is unclear whether this dataset can represent all of the botnet traffic. There can be a possibility of performance drop when tested with different datasets. Specific analysis and justification of the dataset in sections 3 or 4 can show your result more reliable.

2. In Table 3, the F1 score when testing with ViT-BoT marginally drops when considering more packets. In general, the performance has to increase when considering more datas. The analysis of the packets at the back, not only the first few, can further justify this result.

3. This paper only considers experimental settings in controlled environments. It would be interesting to discuss real-world implementation, such as how to implement your framework on a real machine, how it works when implemented, considerations and difficulties, etc.

**Strengths And Weaknesses:**

Strength
+ This paper focuses on detecting encrypted communication and shows high performance on both datas.
+ Their experiment is considered in a resource-constrained setup. The result was still high even with reduced data preprocessing time and flow length consideration.
+ Overall the paper is well-structured and easy to understand.

Weakness
- The dataset used for the experiment can be biased. Only 10% of a single dataset, the BoT-IoT dataset, was considered.

---

### Review · Reviewer_dwg6 · 2025-04-14

**Summary Of Contributions:**

This paper presents a method for botnet detection on IoT and edge devices using computer vision methods. The core idea is to convert network traffic (encrypted or not) into RGB images, which are then classified using a Vision Transformer (ViT) model. The model is pretrained on natural images (publicly available foundation models) and fine-tuned on a dataset encompassing 11 classes of benign or malicious traffic. The proposed method demonstrates good accuracy and lower latency compared to selected baselines. To improve performance on edge devices, the authors apply TensorRT optimizations; the optimized model is deployed to an NVIDIA Jetson Orin Nano gateway.

**Audience:**

No

**Broader Impact Concerns:**

I have no concerns.

**Claims And Evidence:**

No

**Requested Changes:**

To improve the quality and relevance of the paper, I recommend the following:

* Expand the literature review to include state-of-the-art and closely related image-based methods for network traffic and malware classification.
* Clarify novelty in relation to related work.
* Add recent and stronger baselines to the experimental section.
* Analyze the contribution of header vs. payload data—especially under encryption—to the model's decisions. This is essential for interpreting the results and understanding what the model actually learns.
* Include additional datasets, such as MalNet or others cited in relevant literature.

**Strengths And Weaknesses:**

# Strengths
* The method performs well on the BoT-IoT dataset, showing its potential for practical deployment.
* The integration of TensorRT and deployment on a Jetson board adds practical relevance and demonstrates real-world applicability.

# Weaknesses
* Limited novelty: the core idea—converting network traffic to images and using neural networks to classify them—is not new, both in general and in particular for cybersecurity tasks, such as malware classification and intrusion detection. This has become a common technique in the field, with open-source tools (e.g., Binvis) supporting flow-to-image conversions. I am including references to similar papers below.

* Incomplete related work: the paper lacks a thorough review of prior art. It omits several relevant and recent references, including works that use near-identical pipelines and should have been used as baselines in the experiments.

* Supported claims: the paper does not explain how the model is able to classify encrypted packets. Since encrypted payloads should be statistically indistinguishable from random data, it's likely that the model relies solely on unencrypted headers. This hypothesis is not tested or discussed, despite the claim that similar accuracy is achieved for both encrypted and unencrypted traffic.

* Interest for TMLR: the paper is mainly a cybersecurity application of machine learning and does not contribute novel machine learning methodology. The model architecture, data preprocessing, training approach, and pipeline are direct applications of standard practices. As such, the work may be more suitable for a cybersecurity venue than a machine learning research journal like TMLR.

* [Minor] Experimental protocol: I was unable to find details on how the data split was performed to obtain the training and test splits.

* [Minor] The inclusion of training scores and loss values is not meaningful for readers. These metrics are typically used for internal validation and should not be presented in a research paper without a compelling reason.

# References
- [Network Intrusion Detection via Flow-to-Image Conversion and Vision Transformer Classification](https://www.researchgate.net/publication/362818998_Network_intrusion_detection_via_flow-to-image_conversion_and_vision_transformer_classification)
- [VINCENT: Cyber-threat detection through vision transformers and knowledge distillation](https://www.sciencedirect.com/science/article/pii/S0167404824002293)
- [IoT Malware Network Traffic Classification using
Visual Representation and Deep Learning](https://pure.port.ac.uk/ws/portalfiles/portal/23347213/IoT_malware_network_traffic_classification.pdf)
- [Enhanced Image-Based Malware Classification Using Transformer-Based Convolutional Neural Networks (CNNs)](https://www.mdpi.com/2079-9292/13/20/4081)
- [Malware Detection Using Frequency Domain-Based Image Visualization and Deep Learning](https://arxiv.org/abs/2101.10578)
- [Classifying Malware Traffic Using Images and Deep Convolutional Neural Network](https://www.researchgate.net/publication/379941775_Classifying_malware_traffic_using_images_and_deep_convolutional_neural_network)
- [Image-based malware representation approach with EfficientNet convolutional neural networks for effective malware classification](https://www.sciencedirect.com/science/article/abs/pii/S2214212622001570)
- [Malware detection using image representation of malware data and transfer learning](https://www.sciencedirect.com/science/article/abs/pii/S0743731522002118)
- [Cybersecurity Intrusion Detection with Image Classification Model
Using Hilbert Curve](https://www.scitepress.org/Papers/2024/123061/123061.pdf)
- [MalNet: A Large-Scale Image Database of Malicious Software](https://arxiv.org/abs/2102.01072)

---

### Note · Authors · 2025-05-06

I have read and agree with the venue's withdrawal policy on behalf of myself and my co-authors.